# Is it cognitive effort you measure? Comparing three task paradigms to the Need for Cognition scale

**Martin Jensen Mækelæ** [1], **Kristoffer Klevjer**[1], **Andrew Westbrook**[2], **Noah S. Eby**[3], **Rikke Eriksen**[1], **Gerit Pfuhl** [1,4]*

**1** Department of Psychology, UiT–The Arctic University of Norway, Tromsø, Norway, **2** Department of Cognitive, Linguistic, and Psychological Sciences, Brown University, Providence, RI, United States of America, **3** Department of Neurology, University of Washington, Seattle, WA, United States of America, **4** Department of Psychology, Norwegian University of Science and Technology, Trondheim, Norway

* gerit.pfuhl@ntnu.no

**Data Availability Statement:** Data can be found at the Open Science Framework, https://osf.io/dywt4/.

**Funding:** The author(s) received no specific funding for this work.

## Abstract

Measuring individual differences in cognitive effort can be elusive as effort is a function of motivation and ability. We report six studies (N = 663) investigating the relationship of Need for Cognition and working memory capacity with three cognitive effort measures: demand avoidance in the Demand Selection Task, effort discounting measured as the indifference point in the Cognitive Effort Discounting paradigm, and rational reasoning score with items from the heuristic and bias literature. We measured perceived mental effort with the NASA task load index. The three tasks were not correlated with each other (all r's < .1, all p's > .1). Need for Cognition was positively associated with effort discounting (r = .168, p < .001) and rational reasoning (r = .176, p < .001), but not demand avoidance (r = .085, p = .186). Working memory capacity was related to effort discounting (r = .185, p = .004). Higher perceived effort was related to poorer rational reasoning. Our data indicate that two of the tasks are related to Need for Cognition but are also influenced by a participant's working memory capacity. We discuss whether any of the tasks measure cognitive effort.

## Introduction

*Laziness is built deep into our nature (Kahneman, 2011, p. 39)*

People tend to choose the least demanding line of action, famously formulated as the "Law of least work" [1]. Although originally applied to physical effort, it also applies to effort in the cognitive domain [2]. The underlying assumption is that there is a cost associated with cognitive effort [3, 4]. The nature of this cost is uncertain [5] but brain imaging studies have shown that increased cognitive effort reduces activity in the reward network [6–8]. It has been proposed that cognitive effort depends on a cost-benefit analysis to find an optimal balance of expenditure [3, 9–12]. However, cognitive effort is everything but well operationalized [13]. Effort has been described as the use of executive functions, use of attention, workload or computational constraints [13].

**Competing interests:** The authors have declared that no competing interests exist.

Proposed explanations for cognitive effort costs include resource limits and computational costs [4, 14–18], metabolic costs or accumulation of by-products [19], and opportunity costs [4, 12]. Assertions of cognitive effort costs and minimization have been proposed to be implicated in a range of fields e.g., behavioral economics [20, 21], executive functions [22], linguistics [23], and judgment and decision-making [24]. Effort is often inferred from the outcome, i.e., answering intuitively is effortless whereas analytically is effortful. As such, effort is often assumed but rarely validated. Not least because of a missing operationalization and its tight relationship with motivation and cognitive ability. This paper is beyond solving the effort problem [13]. Instead, we present six experiments where we compare three measures of cognitive effort against the benchmark Need for Cognition scale (NCS) and report the subjective task demands of each task with the NASA task load index (N-TLX).

There are well-established individual differences in the willingness to engage in cognitively effortful tasks. Those individual differences can be reliably measured with the Need for Cognition Scale [25, 26]. Still, behavioral paradigms measuring cognitive effort are useful for investigating actualized cognitive effort expenditure, decision-making, developmental trajectories and neural underpinnings. Additionally, concerns about the reliability and validity of self-report motivate the use of behavioral paradigms to complement self-report instruments [27]. Behavioral tasks can be combined with physiological measures and used across the lifespan. Accordingly, a range of tasks have been developed to measure cognitive effort spent in a task. We here focus on cognitive effort, though physical and perceptual effort tasks have been developed too [for a review see e.g., 28, 29].

One strand of research uses computerized tasks for measuring choices between cognitively more or less demanding options. Here, choice patterns are seen as an indication of cognitive effort costs or preferences to avoid cognitive effort [30–34]. Another strand of research gauges typical cognitive effort expenditure by using tasks that require cognitively demanding deliberate processing to answer correctly [35–38]. These approaches differ in numerous ways and show partly opposing results, also when used in clinical samples [32, 39–43]. It is therefore of importance to assess to what degree the paradigms measure the same "cognitive effort" construct.

## Task paradigms for measuring cognitive effort

**Rationality battery.** Task performance on rational reasoning tasks (RQ) is an alternative way of measuring thinking disposition or "cognitive miserliness" [35, 44–46]. Thinking disposition is proposed to be on a spectrum with one end being the preference for using computationally more demanding mechanisms for solving tasks, known as an analytic thinking disposition. On the other end of the spectrum is a preference for cognitive shortcuts, namely an intuitive thinking disposition. An intuitive thinking disposition is prone to rely more on heuristics, which can serve to reduce cognitive effort [24]. Task performance on rational reasoning tasks is proposed to depend on using more cognitively demanding mechanisms and avoiding overreliance on heuristic responses (avoiding "miserly information processing") [47]. Suppression of intuitive but wrong answers requires cognitive control [38]. Individual differences have previously been noticed in tasks measuring deliberate reasoning [48]. Toplak et al. [37] showed that the cognitive reflection task, assesses both the ability and willingness to perform cognitive work. However, recent work has questioned whether normative responding is effortful [49–53]. Performance may depend on cognitive ability, not effort [35–37, 54]. Firstly, normative responding can be as fast as heuristic responding [50, 53]. Secondly, the CRT has been shown to correlate highly with numerical tasks [55] and deliberation and rational thinking are highly correlated with cognitive ability [51]. Note, the rationality battery used here is

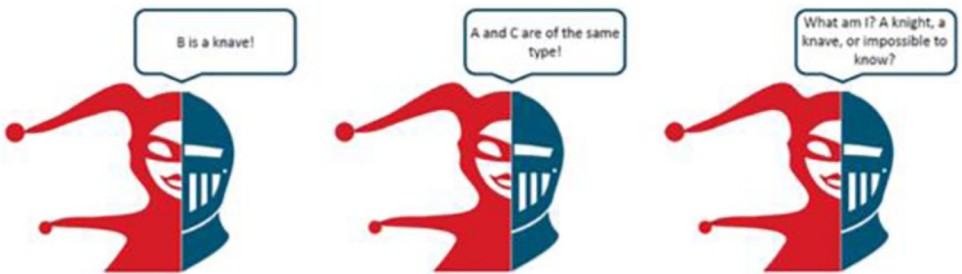

**Fig 1. Example of a task from the rationality battery.** Imagine that there are three inhabitants of a fictitious country, A, B, and C, each of whom is either a knight or a knave. Knights always tell the truth. Knaves always lie. Two people are said to be of the same type if they are both knights or both knaves. A and B make the following statements: A says: "B is a knave." B says: "A and C are of the same type." What is C?.

more than the cognitive reflection test (Fig 1). Such items have been shown to correlate positively with the CRT [37, 56] and the Need for Cognition scale [56]. The CRT has been shown to be positively associated with the Need for Cognition scale too [38], but see [55].

**Demand selection task.** Evidence to support cognitive effort minimization or demand avoidance was shown with the Demand Selection Task paradigm (DST, Fig 2) by Kool, McGuire et al. [30]. In this task, participants make either parity or magnitude judgements for numerical digits. Effort demands are manipulated by the frequency of task shifts: one line of action (high demand) has more frequent task shifts, thus increasing effort demand [57]. DST can be considered an implicit measure of cognitive effort or demand avoidance as participants are not informed of the demands of the tasks or given any incentive to choose high or low demand lines of action. However, several participants detect the demand manipulation, and some evidence suggests this leads to increased effort avoidance [41].

**Cognitive effort discounting paradigm.** Westbrook et al. [32] were able to quantify the individual differences in effort costs with the Cognitive Effort Discounting Paradigm (COGED, Fig 3). In this paradigm, participants make repeated choices between performing a low demand working memory task (1-back) for a small reward or performing a high demand working memory task for a larger reward (n-back, n being 2, 3, 4, 5, or 6). The reward for the low demand task is titrated in response to participants' choices with the aim to find a subjective indifference point between the low demand and high demand option. The COGED thereby quantifies the subjective monetary discounting due to cognitive effort costs across multiple demand levels. Given that task load levels and offer amounts are all explicit, COGED is an explicit cognitive effort measure. Participants experience the effort demand for each load level prior to making choices between explicit monetary offers.

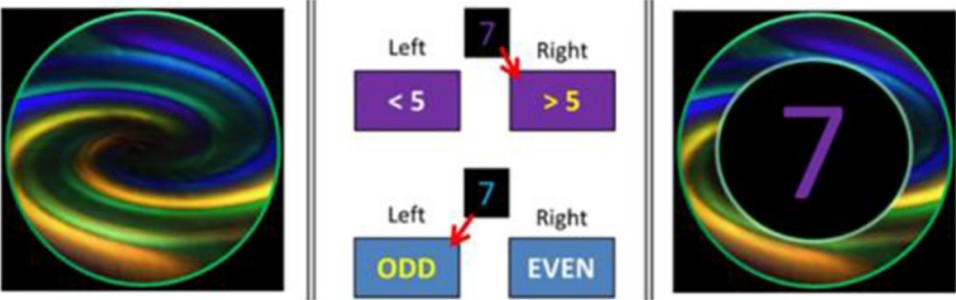

**Fig 2. Schematic illustration of the demand selection task.** In this trial correct responding is by pressing the right mouse button. Note: Participants saw the rules at the beginning and had to remember them during the test blocks.

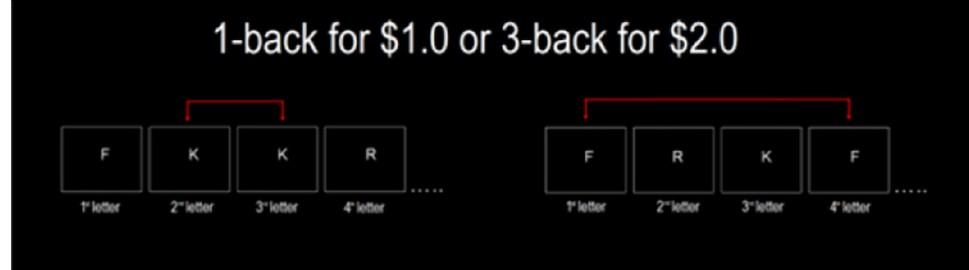

**Fig 3. Schematic illustration of the choice phase of the cognitive effort discounting task.** Note: Participants saw the n-back instructions during training, in the choice phase they were asked to play 1-back vs n-back and the value for 1-back was titrated either up (if n-back chosen) or down (if 1-back chosen).

## The current studies

An outstanding question is whether cognitive effort can reliably be measured. If the three tasks share a common latent construct, that of willingness or propensity to exert cognitive effort, we would expect that all three tasks are related to a measure of enjoying and engaging in cognitively demanding tasks. These individual trait differences in thinking disposition can reliably be measured with the Need for Cognition Scale [25]. The scale has good internal consistency, test-retest reliability, and measurement invariance [26]. People who score high on Need for Cognition (cognizers) seek, evaluate and integrate multiple relevant sources before arriving at an opinion. People who score low on Need for Cognition (cogmisers) tend to use less demanding cognitive processes [25, 58]. Cognizers may value effort whereas cogmisers may avoid effort [59].

If the three behavioral tasks index task-invariant cognitive effort, we expect them to be positively correlated with each other and with the NCS. If the behavioral tasks are not related to each other, the propensity to exert cognitive effort is task-specific. Each task might still be related to the NCS.

Previous work has already shown a positive relationship between effort discounting in the COGED and NCS, and between rational reasoning and NCS [32, 35, 36, 60], but no relationship between demand avoidance in the DST and NCS [42]. It is also not known whether effort discounting is related to rational reasoning and whether cognitive demand avoidance is related to effort discounting and or rational reasoning. Notably, both COGED and DST have been used as measures of cognitive effort in clinical and developmental research [32, 41, 61–64].

We measured test-retest reliability for the DST, COGED and rationality battery. Previously, Stagnaro, Pennycook et al. [65] have shown good test-retest reliability (r = .806) for the cognitive reflection test. Strobel et al. [2020] found questionable test-retest reliability for the DST (ρ = .61). To the best of our knowledge test-retest reliability of the COGED has not been shown.

Finally, subjective effort, which may deviate from objective effort [e.g., 66], can be assessed with the NASA task load index (N-TLX) [67]. Westbrook et al. [32] found increasing subjective ratings of mental and physical effort, temporal demand, failure rate, effort demand and frustration for increasing working memory load levels in the COGED but did not assess it for the choice phase. Here, we report six studies from two independent labs investigating the relationship between demand avoidance in the DST, effort discounting in the COGED, rational reasoning score, NCS and N-TLX. We controlled for working memory capacity by using the n-back performance assessed in the practice phase of the COGED. We also assessed test-retest reliability for the rationality battery, DST and COGED. We report two-sided and non-corrected p-values per study and the mean effect size based on meta-analysis approach. The

S1 File contains details to methods and results of the studies as well as an alternative analysis and plots.

## Study 1–6

### Participants: Study 1 and 2

All study procedures including informed consent were approved by the Institutional Review Board at Washington University in Saint Louis. The studies were conducted in 2013. Participants provided written informed consent.

In study 1 all participants were undergraduate students at Washington University in St. Louis, USA (N = 76, 49 female). The mean age was 21.43 (range 18 to 32 years). In study 2 participants were 91 undergraduate students (47 female, 35 male) at Washington University in St. Louis. The mean age was 23.62 (range 18 to 40 years). Two participants each in study 1 and 2 were excluded due to very bad performance in the n-back task (negative d'). Final sample size for study 1 is N = 74, and for study 2 is N = 80. There was no missing data.

### Participants: Study 3–6

The studies were approved by the institutional review board at the Department of Psychology, UiT–The Arctic University of Norway. The studies were conducted in 2018–2022. All participants provided written informed consent on paper and informed consent (online study 6), respectively.

In study 3 participants were 102 (62 female, 25 male, 15 unknown) undergraduate psychology students at UiT–The Arctic University of Norway and testing was over two sessions. 65 completed both sessions, 82 completed the NCS and the rational reasoning battery, 78 completed the COGED and rational reasoning battery, 63 completed the NCS and COGED. The mean age was 22.6 (range 20 to 38 years).

In study 4 we recruited 40 participants (27 female, range 18 to 37 years). 34 were students at UiT–The Arctic University of Norway, three were full-time workers, and three were high school students. All participants completed both testing sessions. One participant performed randomly in the Demand Selection task and was excluded, i.e., N = 39. Another participant had missing data for the rational reasoning battery.

In study 5 all participants were students (non-psychology) at UiT–The Arctic University of Norway (N = 45, 27 female), mean age was 23.35 (range 18 to 37 years). There was no missing data.

In study 6 participants (*M* = 26.64 years) were recruited from two psychology courses at UiT–The Arctic University of Norway (N = 91, 67 female, 22 male, 2 non-binary; range 19 to 38 years) and from Prolific (prolific.co) (N = 227, 113 female, 110 male, 4 non-binary, range 18 to 62 years). Three participants aborted the choice phase in the COGED and were excluded for parts of the data analysis.

### Materials

**Effort discounting (alterations across studies in brackets).** The COGED task was administered through E-prime 2.0 (Psychology Software Tools, Inc., Sharpsburg, PA) in study 1+2 and through Inquisit (Millisecond.com) in study 3–6. The task started with a practice phase of the n-back task [68]. Participants played all load levels for three runs (six levels in study 1 and 2, four levels in study 3–6).

## Study 1 and 2

All runs consisted of 64 items (consonants, presented in Courier New font, font size 24). Items were presented on screen for 1.5 seconds, during which participants could respond. After 1.5 seconds the items were replaced by a fixation cross. The inter-trial interval was 3.5 seconds. Participants were given feedback about % of targets and % of non-targets correct. Feedback of "Good job!" was given if both scores were above 50% or "Please try harder!" if not. From this phase d' was calculated as an index of working memory capacity (see below).

In the discounting procedure participants were offered to play n = 1 for a small reward or n > 1 for a larger reward. Participants were offered six choices for each load level. The amount for the higher offer (n > 1) was always $2. The reward amount for the lower offer (n = 1) started at $1 and was adjusted up if participants chose the high offer and was adjusted down if participants chose the low offer. Each time a choice was made, the reward amount was adjusted to half as much as in the previous choice. After the last choice (six choices in total), the amount was adjusted to $0.015. The final amount was taken as the participants' subjective indifference point. Participants played five load levels and made six choices for each level, yielding 30 choices in total. To ensure choices reflected participants' preferences, they were told that one of the choices would be selected for them to repeat 10 more times and they would be paid for each repetition. Further, they were told that payment was contingent on maintaining effort, but not on performance. Effort would be monitored by "behavioral clues". All participants completed their randomly selected offer four times and were paid the associated amount.

## Study 3–6

The first phase consisted of five runs per n-back level (2, 3, & 4), each run with five target trials (responding would be a hit), and 10+N non-target trials (responding would be a false alarm) in a pseudo-random sequence. Each trial lasted 2.5 s, and in each trial, participants were presented with a stimulus (one of 20 consonants, centered white letters on a black screen, sans-serif font) for 0.5 s, followed by a black screen for 2.0 s, and during the 2 seconds had to either respond (press 'A' on the keyboard) or not to respond. After each run, the participants were presented with a summary feedback of their accuracy, and after the last run on each n-back level they were presented with a level summary. The second phase consisted of the discounting procedure for 1-back vs. 2-back, 1-back vs. 3-back, and 1-back vs. 4-back, presented in a pseudo-random order across participants. Each block had six runs in which the participants chose between a 1-back task or n-back task. The tasks themselves were equal to the n-back task described above. The discounting amounts were identical to study 1 and 2. In study 3 and 5 participants were informed that they would not receive extra money, thereby eliminating external reward as motivation. In study 4 they could earn a bonus on top of the show-up fee. In study 6 performance in the discounting procedure phase had to be at least 80% (previously 80% for 1-back but at least 100% of that from the practice phase for 2-back, 3-back and 4-back) to count as success. Participants could earn vouchers (students) or a bonus (Prolific). The bonus was related to the amount earned in the discounting phase of the task.

The Average Indifference Point (AIP) across all load levels is the cognitive effort discounting measure used for the bi-variate correlations and regression analysis.

## Cognitive demand avoidance

We used an exact replication of Experiment 3 in Kool et al. (2010). The task was administered on a computer, using MatLab 2018a (The MathWorks, MATLAB, Version 9.4, 2018), with Psychophysics Toolbox 3 extension [69–71]. The task starts with a training phase where

participants complete two different tasks. Participants are presented with a number (between 1 and 9, excluding 5). The number can be either blue or yellow. The color of the number signaled the task required on that trial. If the number is blue, participants must decide if the number is higher or lower than 5. If the number is yellow, then participants must decide if the number is odd or even. Participants indicate their choice by clicking on the right or left side of a computer mouse. During the training phase (60 trials), participants received feedback on their performance. None of the participants had to redo the training phase. In the main task, participants see two colorful balls on screen (they appear along an invisible circle at an angular distance of 45 degrees). The location of the balls changes between runs but is stable throughout a run. Participants must sample from each option but are told they can stay with one if they develop a preference. There are eight runs with 75 trials in each run (600 in total). There is one high demand option (ball) where the task switches with a probability of 0.9, and there is a low demand option where the probability of task switch is 0.1. Task instructions were available in paper format in case participants forgot the rules. Demand avoidance is quantified in terms of the proportion selection of the high demand decks (ball)–thus a demand avoidant participant would score between 0 and .5 and preferring the low demand deck, respectively.

In study 6 the abridged version, i.e., four rounds and 300 trials in total, was used [61].

## Rational reasoning

In study 2 we used the 18 items scale from Toplak, West [37]. This scale includes the original 3-item Cognitive Reflection Test, measuring individual differences in detecting errors and overriding an initial intuitive response [48]. The remaining 15 items were problems from the heuristics and biases literature: two-sample size problems, two gambler's fallacy problems, regression to the mean, a base rate problem, a covariation detection problem, one Bayesian reasoning problem, one conjunction fallacy problem, a denominator neglect problem, a methodological reasoning problem, a probability matching problem, a sunk cost fallacy problem, one outcome bias problem, and a framing problem. Correct answers were scored as 1, incorrect as 0. Total composite score, the rationality quotient RQ, ranged between 0 and 18.

In study 3, 4 and 5 we used 14 items from the heuristics and biases literature. We used items 2–7 from the Cognitive Reflection Test [35], one fully disjunctive reasoning problem "the marriage problem" [72], one probability matching task [73], one probability estimation task "the bus problem" [74], one making sense of medical results problem [75], one Bayesian reasoning problem [76], one covariation detection problem [77], one knight and knave problem [78], one conditional reasoning problem [79]. Correct answers were scored as 1, incorrect as 0. Total composite score ranged between 0 and 14.

In study 6 we used 12 items. These were items 2–7 from the Cognitive Reflection Test [35], one fully disjunctive reasoning problem, "the marriage problem" [72], one knight and knave problem [78], one conditional reasoning problem [79], one covariation problem [37], one base rate problem [36], one making sense of medical results problem [75]. We calculated the proportion of correct items, i.e., the score ranged from 0 (no item correct) to 1 (all items correct).

*Thinking disposition* was measured with the 18-item Need for Cognition Scale (NCS) [80]. An example item is *"I prefer complex to simple problems"*. The 18 items are rated on a 5-point Likert scale from 1 = *"Extremely uncharacteristic of me"* to 5 = *"Extremely characteristic of me"*. Total score range is from 18 to 90.

*Working memory capacity* was measured with the d' from the n-back portion of the COGED task. Here, responding to a previously seen stimulus at the n-th position is a hit, not responding is a miss. Responding too early or too late is a false alarm, not responding to an incorrect letter is a correct rejection. We calculated d' from signal detection theory, i.e., d' = z

(H)–z(FA) where z(H) and z(FA) are the z transforms of hit rate and false alarm rate, respectively. The larger d' the better is a participant's working memory capacity. Study 1 and 2: In the O-Span task participants have to remember sequentially presented words and solve simple, interspersed math problems. The length of the sequence reproduced error-free is used as the maximal working memory capacity.

*Perceived effort*. We asked participants to rate their perceived effort, both mental, physical, temporal, as well as performance, overall effort and frustration using the NASA task load index (N-TLX) [67, 81]. Rating was on a visual analogue scale ranging from 1 = very low to 20 = very high.

## Procedure

Study 1 and 2: Participants were paid 10$ per hour for their participation, and they could earn additional money based on their choice in the COGED. Participants received their payment at the end of the testing session. Testing was completed individually at Washington University in St. Louis.

Study 1: Order of the tasks was: DST, COGED, NCS, O-Span (not reported). Usual participation time was approximately two hours.

Study 2: Order of the tasks was: rational reasoning problems, COGED, NCS, O-Span (not reported). Usual participation time was approximately two hours.

Study 3: Testing took place over two separate sessions in small groups in a computer pool at the campus (UiT). Students took part for course credit and received no monetary compensation. Students could choose to partake in only one session. On day 1, 82 students took part, and were tested on COGED and N-TLX$_{COGED}$. On day 2, approximately 3 weeks later, 84 students took the Rational reasoning items, N-TLX$_{RQ}$, and NCS. 65 students took part in both test sessions. Participants could withdraw or indicate on the consent form that they do not permit to use their data for research, which was once the case. Each session including debriefing and took approximately 1 hour.

Study 4: All participants were tested individually at UiT–The Arctic University of Norway. All participants completed a second testing session between four and eight weeks after the first testing session. Day 1 task order was; DST, Rational reasoning, Bullshit receptivity scale (not included in analysis), NCS, Effort expenditure for rewards task (EEfRT, [82], not included in analysis) and N-TLX$_{EEfRT}$ (not included). Day 2 task order was; DST, NCS, Handgrip [83], not included in analysis), COGED, and N-TLX$_{COGED}$. Participants received a voucher with a fixed amount of 200 NOK (approx. $25) for participation, plus between 50 and 150 NOK depending on task performance in the COGED and EEfRT.

Study 5: Participants were tested individually at UiT–The Arctic University of Norway. All participants received a voucher worth 400 NOK after completing two days of testing (day 2 involved eye tracking, not included here). Each testing session lasted approximately between 1.5 and 2 hours. Relevant is only the first test session. Task order for day 1 was: DST, rational reasoning task, NCS, Teleological reasoning (not reported here).

Study 6: The study was done fully online. Participants read the informed consent and then were randomly assigned to one of the six orders for the three tasks (COGED, DST, rationality reasoning). After each task, they filled out the N-TLX. The NCS was always presented at the end (Prolific sample) and the day after for the UiT students. The tasks were implemented in Inquisit (Millisecond.com). The NCS for UiT participants was implemented in

Qualtrics. Duration was 50–60 min. UiT students received course credit and could earn vouchers and Prolific participants at least £8 for participation plus bonus payment for good performance.

None of the studies was preregistered. All analyses are thus exploratory. With the present sample sizes (minimum is N = 402), we were able to detect correlations of at least r = .177 at an alpha level of .05 and power of .95. Data analysis was done in R [84].

## Results across the six studies

Fig 4 presents the descriptive data for rational reasoning (as percentage of maximum score), demand avoidance (choice of high demand option), cognitive effort discounting (average indifference point), NCS, d', and subjective mental effort rating across the six studies (for study 6 N-TLX values have been converted from the 20-point scale to the 100-point scale). Differences between the six studies are reported in S1 File, including a comparison between lab vs online studies.

We performed bi-variate Pearson product-moment correlations per study. Based on the study-wise correlation coefficients we calculated the mean effect size by using the meta for package [85]. The correlation coefficients are shown in Table 1 and the scatterplots per study in S1 File. An alternative analysis using z-scored values and Bayes Factor is provided in the S1 File.

As can be seen from Table 1, we found a significant positive association between NCS and cognitive effort discounting, and between NCS and rational reasoning score. However, there was no association between demand avoidance in the DST and NCS. This is consistent with previous work. Importantly, there were no significant associations between rational reasoning, cognitive effort discounting or demand avoidance. Thus, the results show that none of the three behavioral tasks measuring cognitive effort are related to each other. Rational reasoning and effort discounting were related to working memory capacity (d').

We next performed three linear mixed regressions to assess whether demand avoidance, rational reasoning or cognitive effort discounting (outcome) was predicted by Need for

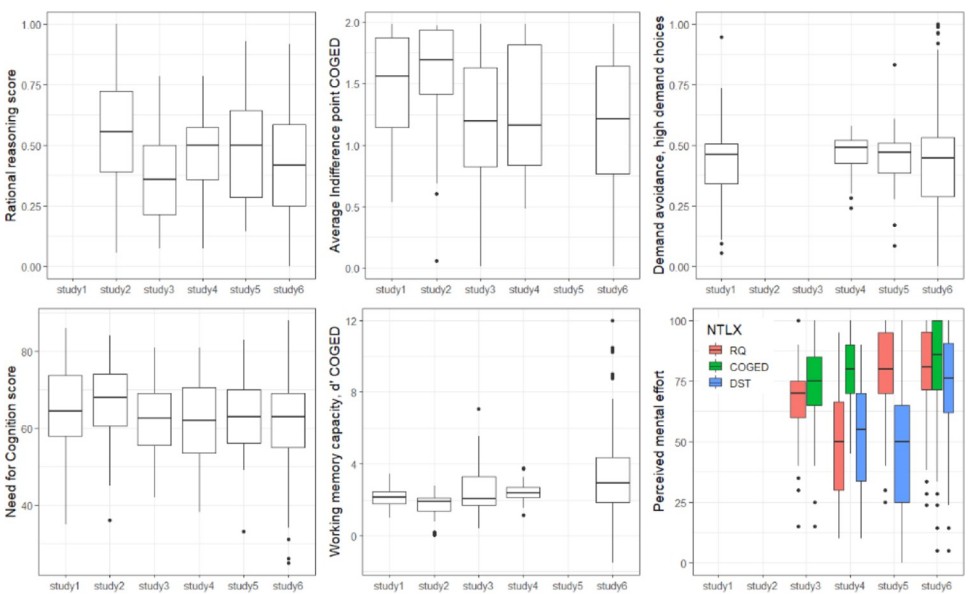

**Fig 4. Descriptive data (box plots) for the main outcome variables per study.**

**Table 1. Pearson's Correlations per study and overall effect size including confidence intervals and p-value.**

| Correlations/ study | Study 1 | Study 2 | Study 3 | Study 4 | Study 5 | Study 6 | mean r | upper CI | lower CI | p |
|---|---|---|---|---|---|---|---|---|---|---|
| NCS–d' | 0.015 | 0.091 | -0.096 | 0.180 | | 0.033 | 0.042 | 0.117 | -0.048 | 0.208 |
| NCS—IP | 0.296 | 0.115 | 0.090 | 0.071 | | 0.177 | 0.168 | 0.248 | 0.086 | <0.001 |
| NCS—High Demand | 0.060 | | | 0.421 | -0.190 | 0.071 | 0.085 | 0.265 | -0.101 | 0.186 |
| NCS—RQ | | 0.171 | 0.007 | 0.405 | 0.225 | 0.184 | 0.176 | 0.270 | 0.079 | <0.001 |
| d'—IP | 0.075 | 0.411 | 0.094 | 0.286 | | 0.107 | 0.185 | 0.313 | 0.050 | 0.004 |
| d'—High demand | 0.173 | | | 0.106 | | -0.043 | 0.040 | 0.184 | -0.107 | 0.298 |
| d'—RQ | | 0.501 | 0.208 | 0.226 | | 0.257 | 0.304 | 0.429 | 0.166 | <0.001 |
| IP—High demand | 0.074 | | | 0.258 | | 0.038 | 0.063 | 0.158 | -0.033 | 0.098 |
| IP—RQ | | 0.192 | -0.002 | -0.016 | | 0.067 | 0.070 | 0.157 | -0.017 | 0.058 |
| High demand—RQ | | | | 0.248 | 0.076 | 0.007 | 0.037 | 0.15 | -0.062 | 0.233 |

Cognition, working memory capacity, any of the other two tasks, or perceived mental effort from the N-TLX, i.e., in each of the three tasks there were five predictors and data was nested within study.

For rational reasoning the fixed effects NCS (t(353) = 3.681, p = .0003), d' (t(353) = 4.797, p < .001) and perceived mental effort (t(353) = -2.518, p = .0124) were significant. Thus, the less effortful the task was perceived, the higher the NCS and the better working memory capacity, the higher was the rational reasoning score.

For demand avoidance none of the fixed effects was significant (all p's > .128). Thus, neither NCS, working memory capacity, perceived mental effort, rational reasoning score or effort discounting were related to the proportion of high demand choices.

For cognitive effort discounting the fixed effect NCS (t(353) = 3.016, p = .0028) was significant. Working memory capacity was not (p = .051), nor any of the other three predictors.

## Reliability

In study 4 we assessed test-retest reliability of the DST. Demand avoidance showed acceptable internal consistency on Day 1 (Cronbach's α = .71), but poor internal consistency on Day 2 (Cronbach's α = .52), and poor reliability across the two testing sessions (r = 0.537, $p < 0.001$).

In two adjacent experiments (see S1 File) we assessed test-retest reliability of a rationality battery (similar items to study 3–6) and the choice pattern in the COGED (average indifference points). Test-retest reliability for the rationality items was good, i.e., Pearson's r(83) = 0.70. Test-retest reliability for effort discounting was good, r(25) = 0.804 (average over three test sessions, session 1 with 2: r = 0.789, session 2 with 3: r = 0.819).

## General discussion

In six studies we did not find that two common cognitive effort tasks, COGED and DST, as well as items from the problem solving and reasoning literature (rationality battery) were related to each other. However, the COGED and the rational reasoning score were positively correlated with the Need for Cognition score and working memory capacity. This was not the case for the Demand Selection task.

COGED is an explicit incentivized choice task between n-back levels. In contrast, the DST requires detecting which of the decks has fewer rule changes and thereby less effortful. Rational reasoning is not necessarily effortful for participants with high cognitive ability [53]. Thus, there are good reasons which may explain why the three tasks do not relate to each other. Still, of the three tasks the COGED and rational reasoning score had positive correlations with

Need for Cognition, i.e., less effort discounting and better rational reasoning was associated with a thinking disposition for engaging in cognitively effortful tasks, replicating previous studies [32, 35, 36, 60]. Relatedly, better working memory, i.e., how good a participant is in the n-back task, was associated with less effort discounting and also a higher rational reasoning score. The latter is in line with a range of studies finding that cognitive ability predicts performance in heuristic and bias tasks [36, 50]. Neither Need for Cognition nor working memory were related to demand avoidance in the DST. This might be due to the implicit nature of the task or perceptual preferences [30]. Note though, that on average participants did prefer the low demand deck, showing that the task does capture a preference for demand avoidance (Fig 4).

Interestingly, and in line with the smart intuitor account, participants scoring well on the rational reasoning items perceived the task as less effortful. Less effort discounting leads to performing harder n-back trials in the task, i.e., task demand becomes higher, thus participants engaging in least effort discounting perceived the task as more effortful. There was no relation between perceived effort and demand avoidance in the DST.

Regarding test-retest reliability, the demand selection task was not reliable, replicating Strobel, Wieder et al. [43]. The rationality items were reasonable reliable and the COGED had good test-retest reliability.

## Rational reasoning items may not measure cognitive effort

Rational reasoning tasks have been used as a convenient, fast and implicit measure of successfully engaging in deliberate reasoning. Indeed, those scoring high on the Need for Cognition scale do perform better on these tasks. Remarkable is also its good test-retest reliability [63]. However, a range of studies question the assumption that performing well on those items is effortful [86–88]. Deliberation can still be effortful, but the items commonly used, also in our studies, may not require deliberation but can be solved by intuition [53]. Researchers should be mindful that performance is dependent on sufficient analytical and reflective abilities yet to be properly defined [89]. Despite task performance being linked to multiple real-world outcomes [90], we caution the use of rational reasoning items to gauge cognitive effort.

## Cognitive effort discounting measures cognitive effort

COGED is a behavioral economic approach to assess cognitive effort discounting of monetary rewards. It is a useful tool for explicitly assessing cognitive effort expenditure and cognitive effort costs. The task was subjectively rated as the most mentally demanding task in our studies. COGED is based on the n-back, a well-established working memory paradigm with parametrically varying cognitive load. Thus, a strength of the COGED paradigm is that performance level is adjusted to a participant's ability and performance in the practice phase. In addition, the measure can provide an estimate of working memory, which is convenient as this allows for correction of cognitive ability which is a confounding variable with most cognitive effort measures. By providing feedback through presenting d' after each round, participant may base their choice of n-back level on this feedback. Participants may prefer levels where they performed better. However, high performing individuals might find the 1-back boring, particularly after engaging in higher levels [91]. This could be mitigated by offering e.g., 2 vs 3-back choice options. COGED might be influenced by individual differences in reward sensitivity, as individuals high in Need for Cognition are less sensitive to rewards [92], but see [7]. This underlines the importance of disentangling intrinsic and extrinsic motivation. Notably, real-world academic achievement was not related to effort discounting in adolescents [93] but

intrinsic motivation assessed with a questionnaire was. We recommend measuring Need for Cognition or Motivation for Cognitive effort [94] to regress out intrinsic motivation.

## Demand avoidance may measure cognitive effort but not reliably

The implicit nature of the DST makes it appealing, however the implicit nature may also limit the tasks' predictive capacity as the task is subject to choices being influenced by factors such as side- and color preferences and also whether or not demand differences were perceived in the first place. In addition, those who detect the demand manipulation show higher demand avoidance compared to those who have not, making the task more into a game to discover the least effortful strategy [30]. The DST showed low test-retest reliability, replicating the finding of [43]. The task was not related to NCS, COGED, or rational reasoning. Given that on average participants did avoid the high demand deck, it is surprising to not find a significant relation with Need for Cognition. For future studies we recommend using a modification of the DST, varying the effort level by changing the frequency of rule changes between rounds [6] and use forced trials to gauge reliable switch costs [30]. Switch costs may index cognitive flexibility and thereby allow to assess relative effort, similar to performance being based on d' in the COGED.

## Limitations and strength

Our samples are mostly students, cautioning generalizability beyond young, healthy, well-educated participants. Since the tasks can feel quite repetitive, a subset of participants might have become bored. We did not inquire about participants' level of boredom. The studies used working memory capacity based on the practice phase in the COGED (n-back task) for individual differences in cognitive abilities. The COGED used in study 1 and 2 does differ from the COGED used in study 3 to 6. We did not measure reward sensitivity or liking of challenges [95]. We limited the comparison to these three tasks, not including the Cognitive Effort Expenditure for Rewards task [96] as we were not aware of the task when starting our studies. This paradigm is an incentivized version of the DST.

Our study is the first to compare three common paradigms for measuring cognitive effort. The results replicated across various samples (psychology undergraduates, non-psychology undergraduates, non-students and students recruited at Prolific) and whether instructions were individually, in groups or solely on screen (online testing). Task-specific effects cannot explain why the three tasks do not relate to each other. However, theoretically, demand avoidance in the DST has to be discovered, rational reasoning has been shown to be intuitive for high performers, and effort discounting is reward sensitive. Future studies should carefully manipulate only one of the aspects the three tasks differ on, to identify which component reflects best individual differences in cognitive effort. Cognitive effort may depend on differences in cognitive ability, intrinsic- and extrinsic motivation, reward sensitivity, task automaticity, and effort costs [5]. Using Thomson and Oppenheimer's framework [13], we have not touched on all levels of analysis, i.e., our studies do not include physiological mechanisms.

## Conclusion

Cognitive effort remains an elusive concept to capture [13]. We did not find that demand avoidance in the DST, cognitive effort discounting in COGED and rational reasoning items measure the same latent construct of cognitive effort. However, both effort discounting and rational reasoning were related to Need for Cognition and working memory capacity. Demand avoidance in the DST had no association with Need for Cognition or any of the other measures. As both DST and COGED are used frequently as measures of cognitive effort including clinical samples, our findings have large implications for interpretations of previous findings.

If the two tasks are measuring different constructs, then findings with one task should not be interpreted as applying to the other task. Lastly, our work highlights the need for developing new behavioral paradigms for measuring cognitive effort [13]. We recommend considering multiple tasks for estimating the latent construct of sensitivity to cognitive effort costs as well as a rating of perceived mental effort.

## Supporting information

**S1 File. Details to the experiments, alternative analysis, figures and reliability.**
(DOCX)

## Acknowledgments

We thank Wouter Kool for providing the Matlab script of the demand selection task. This research did not receive any specific grant from funding agencies in the public, commercial, or not-for-profit sectors.

## Author Contributions

**Conceptualization:** Martin Jensen Mækelæ, Andrew Westbrook, Gerit Pfuhl.

**Data curation:** Kristoffer Klevjer, Andrew Westbrook, Gerit Pfuhl.

**Formal analysis:** Andrew Westbrook, Gerit Pfuhl.

**Funding acquisition:** Andrew Westbrook, Gerit Pfuhl.

**Investigation:** Martin Jensen Mækelæ, Kristoffer Klevjer, Andrew Westbrook, Noah S. Eby, Rikke Eriksen, Gerit Pfuhl.

**Methodology:** Martin Jensen Mækelæ, Kristoffer Klevjer, Andrew Westbrook, Noah S. Eby, Rikke Eriksen, Gerit Pfuhl.

**Project administration:** Andrew Westbrook, Gerit Pfuhl.

**Resources:** Andrew Westbrook, Gerit Pfuhl.

**Software:** Gerit Pfuhl.

**Supervision:** Andrew Westbrook, Gerit Pfuhl.

**Validation:** Gerit Pfuhl.

**Visualization:** Kristoffer Klevjer, Rikke Eriksen, Gerit Pfuhl.

**Writing – original draft:** Martin Jensen Mækelæ, Gerit Pfuhl.

**Writing – review & editing:** Martin Jensen Mækelæ, Kristoffer Klevjer, Andrew Westbrook, Noah S. Eby, Rikke Eriksen, Gerit Pfuhl.

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
