## [Decision Letter · Decision Letter 0]

5 Jun 2023

PONE-D-23-10434

Is it cognitive effort you measure? Comparing three task paradigms to the Need for Cognition scale?

PLOS ONE

Dear Dr. Pfuhl,

Thank you for submitting your manuscript to PLOS ONE. After careful consideration, we feel that it has merit but does not fully meet PLOS ONE’s publication criteria as it currently stands. Therefore, we invite you to submit a revised version of the manuscript that addresses the points raised during the review process.

I and two experts in the field have carefully reviewed your manuscript. We agree that that the study you report on is important and that the paper has promise. One reviewer recommends a minor revision, the other a major revision. I am satisfied that the reviewer concerns can be addressed in a minor revision.

R1 considers the paper to be outstanding, but makes a number of suggestions to improve the paper for its readers. Please address these points.

R2 raises some concerns about the COGED paradigm and I invite you to respond to these and discuss the points raised in your discussion section. R2 also asks for more details in you methods section, sufficient to allow a replication of your analysis. This seems a very reasonable request.

Please also note that this journal does not provide a proofreading service. So please do a careful proof read before submitting a revised version.

We look forward to receiving your revised manuscript.

Kind regards,

Mark Fenton-O'Creevy, PhD

Academic Editor

PLOS ONE

Reviewers' comments:

Reviewer's Responses to Questions

**Comments to the Author**

1. Is the manuscript technically sound, and do the data support the conclusions?

Reviewer #1: Yes

Reviewer #2: Partly

2. Has the statistical analysis been performed appropriately and rigorously? 

Reviewer #1: Yes

Reviewer #2: No

3. Have the authors made all data underlying the findings in their manuscript fully available?

Reviewer #1: Yes

Reviewer #2: Yes

4. Is the manuscript presented in an intelligible fashion and written in standard English?

Reviewer #1: Yes

Reviewer #2: Yes

5. Review Comments to the Author

Reviewer #1: Let's keep it short: This manuscript is outstanding and deserves utmost recognition. Not only does it combine six studies with reasonable sample sizes, not only did the authors use Bayesian statistics in addition to conventional hypothesis testing; not only is the manuscript well-written and easily comprehensible and comes with a thorough and well-balanced discussion of the empirical results: It also provides very important information to the growing number of researchers who investigate cognitive effort. The (non)correlation of established effort-related tasks with each other on the one hand and the correlation of two of these tasks with Need for Cognition on the other hand is per se a finding of great importance and will aid other researchers in planing their studies on effort investment and effort discounting (not only, but also, because the updated effect sizes will allow for appropriate power calculations for future studies). And all this comes with open data and code, which may stimulate re-using these data sets and trying new approaches to examine them.

There are only few remarks from my side:

I missed a section on Statistical Analyses where the software used is cited and the procedures employed are explained. Of course, this can easily be inferred from the *.jasp files, but i f one does not know or use JASP, this information is important to reproduce the results with some other software.

In the context of such a Statistical Analyses section, the power issue should be addressed (e.g., "with the presents sample sizes, we were able to detect correlations of r = .xx - .yy at an alpha level of .05 and a power of .80")

Also, I suggest to add codebooks to the data on the OSF repository. Also, some README file on how to best re-use these data (including where to download JASP) might be helpful to get the most out of these great data sets.

As one final remark: The authors might want to discuss or at least cite the study of Kramer et al. (2021), https://doi.org/10.1016/j.cogdev.2020.100978 who also investigated the relation of the COG--ED paradigm and Need for Cognition.

Overall, I regard this manuscript as an extremely important contribution the field of research on effort discounting including individual differences in this regard. I am happy that I was asked to review this manuscript (so I already now know about these important findings) and hope to see it published very soon (so that others will know about these finding, too).

Signed Review,

Alex Strobel

Reviewer #2: This manuscript describes the results of an analysis of three different cognitive effort tasks across six different studies. The research question is justified and relevant. The introduction and discussion are well written. I have two major points of criticism, one regarding the validity of the conclusions on COGED and Need for Cognition, and one on the level of detail, as well as several minor points.

Major points:

1. The COGED paradigm assumes that all participants inherently prefer the easiest level. It assigns the highest subjective value to 1-back and compares each more difficult level with 1-back in a fashion that does not allow the subjective value for any other level to exceed that of 1-back. This is highly problematic for drawing conclusions about associations with traits such as Need for Cognition. Since Need for Cognition describes the tendency to seek out and enjoy effortful cognitive activities, it would be misguided to assume that participants high in Need for Cognition prefer 1-back over any other level. 1-back is so monotonous and has such a low cognitive demand that it is highly aversive to those who enjoy effort. The subjective values returned by the COGED paradigm are therefore only able to reflect the true preference pattern of anyone who does indeed prefer 1-back over any other level. The “average indifference points” that were used here as an operationalisation of effort discounting are confounded by this distortion: Since the subjective values of participants with high Need for Cognition are artificially high for 1-back, even though they might not like 1-back at all, the “average indifference points” are also artificially high. Seen as a curve of subjective values across n-back levels, the area under the curve might in reality not even be larger for those with high Need for Cognition, it’s just that the peak of the curve is shifted to the right for them. There is a preprint that raises that concern and replicates the findings of Westbrook et al. (2013) with an adaptation of the COGED paradigm (Zerna, J., Scheffel, C., Kührt, C., & Strobel, A. (2022, March 24). When easy is not preferred: A discounting paradigm to assess load-independent task preference. https://doi.org/10.31234/osf.io/ysh3q). The results of the present manuscript should be interpreted with substantial amounts of caution when it comes to individual differences and traits like Need for Cognition.

2. The level of detail in reporting the methods is not sufficient. Especially since the results were not preregistered (or preregistration has not been stated in the manuscript), the description of the analyses is lacking clarity and specificity. A replication would not be possible with the current state of the methods/results section. I highly recommend that the authors provide explicit information in the methods/results section regarding outlier exclusion, handling of missing data, violations of distribution assumptions, transformations, formulas to compute scores such as d’ and subjective values, control variables, DVs and IV in the regressions, estimators, potential confounds, etc.

Minor points:

1. It seems like the authors accidentally added a question mark at the end of the title during submission. The title in the pdf is correct, but the manuscript name contains the question mark.

2. I do not think that stating the results in lines 62 to 66 is helpful for the reading flow. The authors might want to consider removing or replacing these lines with a very brief overview of the study design, e.g. population, online or lab, between or within.

3. Line 79, line 151, line 258, line 400+6 (unlabelled), and line 431 contain numbers for in-text citations, please replace them with the author names of the corresponding reference.

4. The grammar in line 80—81 is off (“However, recent work has questioned […] and propose […]”).

5. The brackets don’t add up in line 88 and in line 130.

6. The original paper on the COGED paradigm by Westbrook et al. (2013) referred to the levels by different colours rather than by their names to avoid anchoring effects in the effort discounting. Please justify briefly why you deviated from this method.

7. The introduction mentions the challenge of measuring effort, but remains quite superficial in its criticism. I recommend including insights from the following paper: Thomson, K. S., & Oppenheimer, D. M. (2022). The "Effort Elephant" in the Room: What Is Effort, Anyway?. Perspectives on psychological science : A journal of the Association for Psychological Science, 17(6), 1633–1652. https://doi.org/10.1177/17456916211064896

8. The description of people high in Need for Cognition as people who “engage with information to make sense of things and events” on lines 132—133 is very colloquial. The authors might want to consider changing this phrase and briefly explaining the different ways in which the elaborate information processing style affects task performance, academic achievement, etc.

9. The “than” in line 137 should be a “then”, I believe.

10. The citation of Strobel et al. (2020) in line 141 is not in the number-citation style.

11. Lines 146—149 contain four different versions of where to put the hyphen in “test-retest reliability”, please be consistent throughout the manuscript.

12. I believe the second half of the sentence is missing in line 164.

13. The reported age parameters are not consistent between studies on page 8. For some studies, the authors report mean and standard deviation, for some only the range. Please indicate whether this information is not available for these studies or, if it is, you might want to consider presenting the sample characteristics in tabular rather than text form. That would save room and make it easier to draw comparisons between samples.

14. Since some of the included studies are 10 years old, I assume that there are other publications based on the data of those studies. Please indicate those as references in the descriptions of the studies on page 8.

15. The stimulus durations of the n-back task differ greatly between studies (1.5 seconds in study 1 and 2 versus 0.5 seconds in study 3 to 6), as do the response time windows (1.5 seconds in study 1 and 2 versus 2 seconds in study 3 to 6) and response patterns (not reported for study 1 and 2 versus pressing one key for a target in study 3 to 6). Please mention briefly in what way these differences might influence the comparability between studies and whether or not you incorporated this into the analyses.

16. The methods section does not describe how the subjective values are computed. Please add this information.

17. Please include the exact formula of how you computed d’.

18. The NASA-TLX in its analogue form is measured on a scale from 1 to 20, because each line indicates a step of 5 points, 100 in total. If “21” is a typo, please correct it. If not, please justify why you deviated from the original NASA-TLX scaling.

19. If you have preregistered your analyses, please include a link to the time-stamped preregistration in the manuscript. I you have not, please justify why, and note that the analyses are exploratory rather than confirmatory.

20. Please increase the font size in figure 4, it is impossible to read.

21. The formatting of “BF10” is inconsistent throughout the manuscript, sometimes subscript, sometimes not. Please make sure it is consistent.

22. The BF10 in the fourth row of table 1 should be written in bold.

23. Please be more specific in the description of your analyses on line 354—357. These sentences do not sufficiently describe the operationalisations and regression formulas, i.e. which variables were the IV and the DV, which variables were controlled for, etc. The descriptions of the results on the following lines should be adjusted accordingly.

24. Since participants received feedback on their accuracy after each n-back level, the association of accuracy and effort discounting might be confounded by the fact that participants used the feedback as an anchor during effort discounting, preferring levels in which they knew they performed well. Please discuss this possibility briefly in the discussion section.

25. It is fantastic that the authors share their data and analysis scripts. However, looking into the subfolders of the studies on OSF, I think other researchers will find it hard to navigate the csv-files based on their name. Some names contain an author name and a number, others only construct acronyms, which then reappear in multiple file names with other constructs. A clear naming convention would be helpful, as well as meta-data (tags) to increase the discoverability of the project.

6. PLOS authors have the option to publish the peer review history of their article (what does this mean?). If published, this will include your full peer review and any attached files.

Reviewer #1: **Yes: **Alexander Strobel

Reviewer #2: No

---

## [Author Response · Author response to Decision Letter 0]

18 Jul 2023

Reviewer #1: Let's keep it short: This manuscript is outstanding and deserves utmost recognition. Not only does it combine six studies with reasonable sample sizes, not only did the authors use Bayesian statistics in addition to conventional hypothesis testing; not only is the manuscript well-written and easily comprehensible and comes with a thorough and well-balanced discussion of the empirical results: It also provides very important information to the growing number of researchers who investigate cognitive effort. The (non)correlation of established effort-related tasks with each other on the one hand and the correlation of two of these tasks with Need for Cognition on the other hand is per se a finding of great importance and will aid other researchers in planing their studies on effort investment and effort discounting (not only, but also, because the updated effect sizes will allow for appropriate power calculations for future studies). And all this comes with open data and code, which may stimulate re-using these data sets and trying new approaches to examine them.

There are only few remarks from my side:

I missed a section on Statistical Analyses where the software used is cited and the procedures employed are explained. Of course, this can easily be inferred from the *.jasp files, but i f one does not know or use JASP, this information is important to reproduce the results with some other software.

We thank you for your kind words and have now updated the method and analysis section. We have used R and the metafor package. The change in analysis method is due to a re-evaluation of z-scoring the values and pooling over the six studies (this analysis has been moved to the supplementary material). The meta-analytical approach allows us to include the correlations from each study and weighting their contribution by the sample size. This is more appropriate given some methodological differences between the studies. Nevertheless, the overall results have not changed, and the codebook includes the code to z-score the data, and the supplementary material section comparing online vs lab studies use the z-scored values.

In the context of such a Statistical Analyses section, the power issue should be addressed (e.g., "with the presents sample sizes, we were able to detect correlations of r = .xx - .yy at an alpha level of .05 and a power of .80")

We now use a meta-analytical approach, but for your information, for the bivariate correlations with N=402, a power of 95%, alpha = 5% we would be able to detect a small effect of r = .18 (G*power 3.1, two-sided testing). For N=591 an effect as small as r = .15 would have been possible.

Also, I suggest to add codebooks to the data on the OSF repository. Also, some README file on how to best re-use these data (including where to download JASP) might be helpful to get the most out of these great data sets.

Thank you for this suggestion. We now provide an R file with notes serving as codebook. All data has been collated in an excel file (separate sheets) and named study 1 to 6. Information about the studies are provided in the supplementary file and the codebook. 

As one final remark: The authors might want to discuss or at least cite the study of Kramer et al. (2021), https://doi.org/10.1016/j.cogdev.2020.100978 who also investigated the relation of the COG--ED paradigm and Need for Cognition.

Thank you for this interesting paper, I enjoyed very much reading it and was not aware of it beforehand. I find it intriguing that there was no association with academic achievement. We briefly mention this in our discussion.

Overall, I regard this manuscript as an extremely important contribution the field of research on effort discounting including individual differences in this regard. I am happy that I was asked to review this manuscript (so I already now know about these important findings) and hope to see it published very soon (so that others will know about these finding, too).

Signed Review,

Alex Strobel

Reviewer #2: This manuscript describes the results of an analysis of three different cognitive effort tasks across six different studies. The research question is justified and relevant. The introduction and discussion are well written. I have two major points of criticism, one regarding the validity of the conclusions on COGED and Need for Cognition, and one on the level of detail, as well as several minor points.

Major points:

1. The COGED paradigm assumes that all participants inherently prefer the easiest level. It assigns the highest subjective value to 1-back and compares each more difficult level with 1-back in a fashion that does not allow the subjective value for any other level to exceed that of 1-back. This is highly problematic for drawing conclusions about associations with traits such as Need for Cognition. Since Need for Cognition describes the tendency to seek out and enjoy effortful cognitive activities, it would be misguided to assume that participants high in Need for Cognition prefer 1-back over any other level. 1-back is so monotonous and has such a low cognitive demand that it is highly aversive to those who enjoy effort. The subjective values returned by the COGED paradigm are therefore only able to reflect the true preference pattern of anyone who does indeed prefer 1-back over any other level. The “average indifference points” that were used here as an operationalisation of effort discounting are confounded by this distortion: Since the subjective values of participants with high Need for Cognition are artificially high for 1-back, even though they might not like 1-back at all, the “average indifference points” are also artificially high. Seen as a curve of subjective values across n-back levels, the area under the curve might in reality not even be larger for those with high Need for Cognition, it’s just that the peak of the curve is shifted to the right for them. There is a preprint that raises that concern and replicates the findings of Westbrook et al. (2013) with an adaptation of the COGED paradigm (Zerna, J., Scheffel, C., Kührt, C., & Strobel, A. (2022, March 24). When easy is not preferred: A discounting paradigm to assess load-independent task preference. https://doi.org/10.31234/osf.io/ysh3q). The results of the present manuscript should be interpreted with substantial amounts of caution when it comes to individual differences and traits like Need for Cognition.

Thank you very much for your constructive feedback. I attended a conference where the Registered Report had been presented. This “boring” feeling of playing 1-back after having played 4-, 5- or 6-back aligns well with personal experience. However, we did not find a non-linear relationship between the indifference points for 1 vs 2, 1 vs 3 or 1 vs 4 at a group-level. For example, in study 3 the average IP for 1 vs 2 was 1.39, for 1 vs 3 was 1.2 and for 1 vs 4 back it was .94. As Zerna et al. do, to test it one would need to offer 2 vs 3 and 2 vs 4 back (and 3 vs 4 back). However, this would – in our opinion – unnecessarily complicate the comparison to the standard COGED paradigm. We have mentioned this opportunity and suggest further research into this exciting topic. 

2. The level of detail in reporting the methods is not sufficient. Especially since the results were not preregistered (or preregistration has not been stated in the manuscript), the description of the analyses is lacking clarity and specificity. A replication would not be possible with the current state of the methods/results section. I highly recommend that the authors provide explicit information in the methods/results section regarding outlier exclusion, handling of missing data, violations of distribution assumptions, transformations, formulas to compute scores such as d’ and subjective values, control variables, DVs and IV in the regressions, estimators, potential confounds, etc.

We have amended the methods section to report exclusions. Since our main analysis is based on bivariate correlations missing data was listwise excluded but no participant was excluded if they had missing data in one test. We report the formula for d’. We did not use the subjective value, thanks for catching that, solely the indifference point (the SV and IP correlate highly). We have updated the section on the regression, using now a general linear model to align better with the nested structure of our data (we use a meta-analytical approach, not z-scoring).

Based on your suggestions we have changed to a meta-analytical approach, using the metafor package in R. This avoids any data transformation as the bivariate correlations from each study and the sample size are sued to calculate an overall effect size. 

Minor points:

1. It seems like the authors accidentally added a question mark at the end of the title during submission. The title in the pdf is correct, but the manuscript name contains the question mark.

2. I do not think that stating the results in lines 62 to 66 is helpful for the reading flow. The authors might want to consider removing or replacing these lines with a very brief overview of the study design, e.g. population, online or lab, between or within. 

Thank you, we have fixed that by rewriting the section.

3. Line 79, line 151, line 258, line 400+6 (unlabelled), and line 431 contain numbers for in-text citations, please replace them with the author names of the corresponding reference. 

Thanks for pointing that out, we have done our best to persuade the reference software to comply. 

4. The grammar in line 80—81 is off (“However, recent work has questioned […] and propose […]”). 

The sentence got rewritten. Thanks

5. The brackets don’t add up in line 88 and in line 130. 

Thanks, we have carefully checked all bracket issues.

6. The original paper on the COGED paradigm by Westbrook et al. (2013) referred to the levels by different colours rather than by their names to avoid anchoring effects in the effort discounting. Please justify briefly why you deviated from this method.

There was no deep theoretical reason why we switched from using color labels to vowel labels, other than we thought the vowels would provide a bit more of a structured heuristic to help people recall the ordering of levels by relative demand. The primary reason for switching from 1.5 s inter-stimulus intervals to 2 s intervals in the N-back was to reduce the time pressure so that levels would be distinguished by cognitive load rather than urgency / time pressure There was not much of an effect on performance (see Figure 4).

7. The introduction mentions the challenge of measuring effort, but remains quite superficial in its criticism. I recommend including insights from the following paper: Thomson, K. S., & Oppenheimer, D. M. (2022). The "Effort Elephant" in the Room: What Is Effort, Anyway?. Perspectives on psychological science : A journal of the Association for Psychological Science, 17(6), 1633–1652. https://doi.org/10.1177/17456916211064896

Thank you for this excellent paper, I really enjoyed reading it. We have amended our introduction to highlight some of the key points from this paper. 

8. The description of people high in Need for Cognition as people who “engage with information to make sense of things and events” on lines 132—133 is very colloquial. The authors might want to consider changing this phrase and briefly explaining the different ways in which the elaborate information processing style affects task performance, academic achievement, etc.

We apologize for this and have rewritten the section

9. The “than” in line 137 should be a “then”, I believe.

Rewritten, thanks

10. The citation of Strobel et al. (2020) in line 141 is not in the number-citation style. 

Thank you, this has been corrected.

11. Lines 146—149 contain four different versions of where to put the hyphen in “test-retest reliability”, please be consistent throughout the manuscript.

We apologize for that and have corrected it.

12. I believe the second half of the sentence is missing in line 164.

We have removed it.

13. The reported age parameters are not consistent between studies on page 8. For some studies, the authors report mean and standard deviation, for some only the range. Please indicate whether this information is not available for these studies or, if it is, you might want to consider presenting the sample characteristics in tabular rather than text form. That would save room and make it easier to draw comparisons between samples.

We report the range for all studies as we have this information for all studies. 

14. Since some of the included studies are 10 years old, I assume that there are other publications based on the data of those studies. Please indicate those as references in the descriptions of the studies on page 8.

No, none of the studies has been published previously

15. The stimulus durations of the n-back task differ greatly between studies (1.5 seconds in study 1 and 2 versus 0.5 seconds in study 3 to 6), as do the response time windows (1.5 seconds in study 1 and 2 versus 2 seconds in study 3 to 6) and response patterns (not reported for study 1 and 2 versus pressing one key for a target in study 3 to 6). Please mention briefly in what way these differences might influence the comparability between studies and whether or not you incorporated this into the analyses.

Thank you for raising this point. Based on this valid concern we have used a meta-analytical approach. This approach uses the study-wise bivariate correlations and weights them (based on the sample size). This takes care of the methodological differences in the COGED. We plot in the supplementary material the bivariate correlations per study. As can be seen from Figure S1 to S10, there was no systematic difference between the first two studies using the E-prime implementation of COGED and studies 3-6 using the Inquisit implementation of COGED.

16. The methods section does not describe how the subjective values are computed. Please add this information.

This was a glimpse from our side, we did not calculate the SV, only the indifference point. 

17. Please include the exact formula of how you computed d’.

We have amended how d’ is calculated.

18. The NASA-TLX in its analogue form is measured on a scale from 1 to 20, because each line indicates a step of 5 points, 100 in total. If “21” is a typo, please correct it. If not, please justify why you deviated from the original NASA-TLX scaling.

The original N-TLX assesses work load on five 7-point scales. Increments of high, medium and low estimates for each point result in 21 gradations on the scales (see their Figure 8.6).

19. If you have preregistered your analyses, please include a link to the time-stamped preregistration in the manuscript. I you have not, please justify why, and note that the analyses are exploratory rather than confirmatory.

We have clarified that the collated analysis is exploratory. 

20. Please increase the font size in figure 4, it is impossible to read.

We apologize for this and have increased the font size

21. The formatting of “BF10” is inconsistent throughout the manuscript, sometimes subscript, sometimes not. Please make sure it is consistent.

We provide this analysis in the supplementary material and have corrected the BF10 issue.

22. The BF10 in the fourth row of table 1 should be written in bold.

We disagree, the BF = 1.74 is < 3 but > .3 and hence inconclusive.

23. Please be more specific in the description of your analyses on line 354—357. These sentences do not sufficiently describe the operationalisations and regression formulas, i.e. which variables were the IV and the DV, which variables were controlled for, etc. The descriptions of the results on the following lines should be adjusted accordingly.

Thank you, we have amended the description. Please note that we now use a general linear model since data is nested within the studies. The formula is, e.g., 

RQ ~ NCS +d’ + COGED + DST_HD + mental effort N-TLX + (1|study)

24. Since participants received feedback on their accuracy after each n-back level, the association of accuracy and effort discounting might be confounded by the fact that participants used the feedback as an anchor during effort discounting, preferring levels in which they knew they performed well. Please discuss this possibility briefly in the discussion section.

We now have explicitly stated this in the discussion. Please note, that this is intentional, i.e., the choice should be based on knowing how well they performed.

25. It is fantastic that the authors share their data and analysis scripts. However, looking into the subfolders of the studies on OSF, I think other researchers will find it hard to navigate the csv-files based on their name. Some names contain an author name and a number, others only construct acronyms, which then reappear in multiple file names with other constructs. A clear naming convention would be helpful, as well as meta-data (tags) to increase the discoverability of the project.

Thank you for raising this non-FAIR issue. We now provide a codebook and xlsx file. For study 1 and 2 we uploaded the raw data too (legally permissible). We have also included tags on OSF.

---

## [Editor Report · Decision Letter 1]

3 Aug 2023

“Is it cognitive effort you measure? Comparing three task paradigms to the Need for Cognition scale”

PONE-D-23-10434R1

Dear Dr. Pfuhl,

Thank you for your careful response to comments from myself and reviewers. I am delighted move this very useful contribution to the study of cognitive effort into the publication process.

We’re pleased to inform you that your manuscript has been judged scientifically suitable for publication and will be formally accepted for publication once it meets all outstanding technical requirements.

Kind regards,

Mark Fenton-O'Creevy, PhD

Academic Editor

PLOS ONE

---

## [Editor Report · Acceptance letter]

8 Aug 2023

PONE-D-23-10434R1 

Is it cognitive effort you measure? Comparing three task paradigms to the Need for Cognition scale 

Dear Dr. Pfuhl:

I'm pleased to inform you that your manuscript has been deemed suitable for publication in PLOS ONE. Congratulations! Your manuscript is now with our production department. 

Kind regards, 

on behalf of

Professor Mark Fenton-O'Creevy 

Academic Editor

PLOS ONE